# Link between Omega 3 Fatty Acids Carried by Lipoproteins and Breast Cancer Severity

**DOI:** 10.3390/nu14122461

**Published:** 2022-06-14

**Authors:** Christine Bobin-Dubigeon, Hassan Nazih, Mikael Croyal, Jean-Marie Bard

**Affiliations:** 1EA 2160—IUML FR3473 CNRS, Nantes Université, UMR6286, US2B, 44035 Nantes, France; 2Department of Biopathology, Institut de Cancérologie de l’Ouest, 44805 Saint-Herblain, France; jean-marie.bard@ico.unicancer.fr; 3CRNHO, West Human Nutrition Research Center, 44000 Nantes, France; el-hassane.nazih@univ-nantes.fr (H.N.); mikael.croyal@univ-nantes.fr (M.C.); 4ISOMer UE2160 IUML, Nantes Université, CNRS3473, 44300 Nantes, France; 5CHU Nantes, Nantes Université, CNRS, Inserm, BioCore, US16, SFR Bonamy, 44000 Nantes, France

**Keywords:** breast cancer, EPA, DHA, lipoproteins, HR−, omega 3 PUFA

## Abstract

According to the International Agency for Research on Cancer (IARC) more than 10% of cancers can be explained by inadequate diet and excess body weight. Breast cancer is the most common cancer affecting women. The goal of our study is to clarify the relationship between ω3 fatty acids (FA) carried by different lipoproteins and breast cancer (BC) severity, according to two approaches: through clinic-biological data and through in vitro breast cancer cell models. The clinical study has been performed in sera from a cohort of BC women (*n* = 140, ICO, France) whose tumors differed by their hormone receptors status (HR− for tumors negative for estrogen receptors and progesterone receptors, HR+ for tumors positive for either estrogen receptors or progesterone receptors) and the level of proliferation markers (Ki-67 ≤ 20% Prolif− and Ki-67 ≥ 30% Prolif+). Lipids and ω3FA have been quantified in whole serum and in apoB-containing lipoproteins (Non-HDL) or free of it (HDL). Differences between Prolif− and Prolif+ were compared by Wilcoxon test in each sub-group HR+ and HR−. Results are expressed as median [25th–75th percentile]. Plasma cholesterol, triglycerides, HDL-cholesterol and Non-HDL cholesterol did not differ between Prolif− and Prolif+ sub-groups of HR− and HR+ patients. Plasma EPA and DHA concentrations did not differ either. In the HR− group, the distribution of EPA and DHA between HDL and Non-HDL differed significantly, as assessed by a higher ratio between the FA concentration in Non-HDL and HDL in Prolif− vs. Prolif+ patients (0.20 [0.15–0.36] vs. 0.04 [0.02–0.08], *p* = 0.0001 for EPA and 0.08 [0.04–0.10] vs. 0.04 [0.01–0.07], *p* = 0.04 for DHA). In this HR− group, a significant increase in Non-HDL EPA concentration was also observed in Prolif− vs. Prolif+ (0.18 [0.13–0.40] vs. 0.05 [0.02–0.07], *p* = 0.001). A relative enrichment on Non-HDL in EPA and DHA was also observed in Prolif− patients vs. Prolif+ patients, as assessed by a higher molar ratio between FA and apoB (0.12 [0.09–0.18] vs. 0.02 [0.01–0.05], *p* < 0.0001 for EPA and 1.00 [0.73–1.69 vs. 0.52 [0.14–1.08], *p* = 0.04 for DHA). These data were partly confirmed by an in vitro approach of proliferation of isolated lipoproteins containing EPA and DHA on MDA-MB-231 (HR−) and MCF-7 (HR+) cell models. Indeed, among all the studied fractions, only the correlation between the EPA concentration of Non-HDL was confirmed in vitro, although with borderline statistical significance (*p* = 0.07), in MDA-MB-231 cells. Non-HDL DHA, in the same cells model was significantly correlated to proliferation (*p* = 0.04). This preliminary study suggests a protective effect on breast cancer proliferation of EPA and DHA carried by apo B-containing lipoproteins (Non-HDL), limited to HR− tumors.

## 1. Introduction

With more than 685,000 deaths per year worldwide, breast cancer remains the most important cause of cancer death for women. Diet and environmental factors play a crucial role in the development of these cancers. Thus, more than 10% of cancers are associated with an unbalanced diet [1].

Fatty acids, especially omega-3 polyunsaturated fatty acids (ω3-PUFA) could influence the severity of cancer. These omega-3 fatty acids constitute a heterogeneous family of which the most biological active are eicosapentaenoic acid or EPA and docosahexaenoic acid or DHA, mainly found in marine fish as an edible source [2]. They are involved in numerous physiological regulation mechanisms and have antioxidant and anti-inflammatory properties. These PUFA are essential for the development of the nervous system, brain and retina. Their effects on health are globally recognized as beneficial [3], justifying the nutritional intake recommendations by international health authorities [4]. Thus, clinical data in the cardiovascular field suggest that a daily intake (2 g/d) of EPA, in patients with high cardiovascular risk, significantly prevents cardiovascular events [5]. Recent focus on SARS-CoV-2 infected patients has recently suggested an optimized ω3-PUFA to prevent infection [6]. The effects of ω3-PUFA intake and breast cancer risk have been studied in different meta-analyses, especially observational studies, but weak association has been identified [7].

Similar to the other fatty acids, EPA and DHA are oxidized in mitochondria. After incorporation into phospholipids, they are incorporated in the cellular membranes or in triglycerides (TG) embedded in lipoproteins excreted into the circulation or stored in adipose tissues.

Circulating lipoproteins are heterogeneous in terms of physical and biological characteristics. They are usually separated into three main classes based on their density: Very Low-Density Lipoproteins (VLDL), Low-Density Lipoproteins (LDL) and High-Density Lipoproteins (HDL). VLDL and LDL share a common protein component, apolipoprotein (apo) B, which allows them to be recognized and internalized by the cellular apo B/E receptor, leading to the delivery of lipids to the cells. By contrast, HDL, through its apo A-I component, plays a central role in reverse cholesterol transport [8]. In clinical and epidemiological studies, circulating lipoproteins are commonly separated into two groups, based on their metabolic behavior, HDL and Non-HDL. EPA and DHA are known to impact lipoprotein metabolism by reducing hepatic VLDL production, inducing a TG-lowering effect, as described in supplementation studies [9,10].

The effects of PUFA intake on the risk of cancers differ according to the PUFA derivatives. The ω6-PUFA derivatives are generally associated with an increase in breast cancer risk by promoting the occurrence, progression, and metastasis of breast tumors [11]. Dietary supplementation of ω6 seems to be associated with an increased risk of breast cancer as recently described in a Japanese epidemiologic nutritional cohort [12]. Conversely, ω3 PUFAs are suspected to have anticancer effects. The link between breast cancer, the main female cancer location, and ω3-PUFA is still not clear [13] even though it has been explored in vivo through supplementation preclinical experiments, and in clinical studies [14,15].

Here, we hypothesized that ω3-PUFA may influence the severity of breast cancer depending on the nature of lipoproteins carrying these FA. The goal of our study was to clarify the relationship between EPA and DHA carried by HDL and Non-HDL and breast cancer (BC). To test this hypothesis, we compared their circulating levels in a cohort of patients with BC, differing by the characteristics of their tumor. In addition, these lipoproteins were tested for their capability to induce proliferation in an in vitro model of BC cells exhibiting (MCF7 cells) or not (MDA cells) the estrogen receptor.

## 2. Materials and Methods

### 2.1. Patients

This study was conducted retrospectively, in a BC cohort before any treatment (surgery or chemotherapy), according to the French legislation (CRB-Tumorothèque–ICO, France- Declaration Number: DC-2018-3321) [16] and written informed consent was obtained for each patient as required by French legislation and the French committee CPP (Comité de Protection des Personnes) for the protection of human rights. The patient cohort has been described in a previous work [17]. Briefly, the anthropometric data (age, weight, height, menopausal status) and the carcinological characteristics (histopronostic grade, the clinical stage, subtypes of breast cancer) were collected.

Patient tumors differed by the expression of estrogen (ER) and/or progesterone (PR) receptor (HR− for ER-PR- and HR+ for ER+ and/or PR+, respectively) and the level of the proliferation marker Ki-67 (Ki-67 ≤ 20%/Prolif− or Ki-67 ≥ 30%/Prolif+). Serum and EDTA plasma samples were collected from patients at the time of diagnosis and before any therapeutic intervention. Among the HR+ group (*n* = 92), 41 tumors were Ki-67 ≥ 30%, and 51 tumors were Ki-67 ≤ 20%. Among the HR− group (*n* = 29 tumors were Ki-67 ≥ 30%, and 19 tumors were Ki-67 ≤ 20%.

### 2.2. Lipid Analysis

Plasma lipid measurements

Total cholesterol, plasma triglycerides, high-density lipoprotein cholesterol (HDL-C) and low-density lipoprotein cholesterol (LDL-C) concentrations were measured using enzymatic kits from Diasys, according to the manufacturer’s instructions (Grabels, France).

HDL and Non-HDL isolation

Apo A-I-containing lipoproteins (HDL) and apoB-containing lipoproteins (Non-HDL) were separated from EDTA plasma (200 µL) by the addition of 2 M MgCl_2_ solution (5 µL) and 4% phosphotungstic acid (20 µL) previously dissolved in 0.15 M sodium hydroxide solution. After centrifugation (30 min, 4 °C, 4000× *g*), supernatants were collected (HDL) and stored at −80 °C until analysis. The pellets (Non-HDL) were washed 2-times using the same procedure. The supernatants were discarded and the pellets were solubilized in 200 µL of 50 mM ammonium bicarbonate (pH 8) and stored at −80 °C until analysis.

For the cell viability test, HDL and non HDL were prepared as follows:

Lipoproteins fractions, high-density lipoproteins [HDL] (density: 1.063 to 1.210 g/mL) and non-high-density lipoproteins [non-HDL], combining VLDL very low-density lipoproteins [VLDL], intermediate-density lipoprotein [IDL] and low-density lipoprotein [LDL] (density: 1.006 to 1.063) were separated from EDTA plasma by density gradient ultracentrifugation as described previously [18,19].

EPA and DHA measurements

EPA and DHA concentrations were determined by liquid chromatography–tandem mass spectrometry (LC-MS/MS) in EDTA plasma, HDL and Non-HDL samples as described previously [20]. Briefly, EPA and DHA were quantified in 100 µL sample aliquots after their extraction by the addition of a methanol/chloroform mixture (2:1; *v*:*v*; Sigma Aldrich^©^, St Quentin Fallavier, France). EPA and DHA were then separated by reversed-phase liquid chromatography before their specific detection by tandem mass spectrometry.

### 2.3. Cell Culture and Viability Assay

The human breast cancer-derived cell lines MCF-7 and MDA-MB-231 were obtained from the European Collection of Animal Cell Cultures (ECACC, Salisbury, UK). Dulbecco’s Modified Eagle’s Medium (DMEM), fetal bovine serum (FBS), glutamine, penicillin–streptomycin, fatty-acid-free bovine serum albumin (BSA), 3-(4,5-Dimethyl-2-thiazolyl)-2,5-diphenyltetrazolium bromide (MTT) and DMSO were provided from Sigma Aldrich (Lyon, France). The MCF-7 and MDA-MB-231 cells were cultured in DMEM supplemented with 10% FBS, 1% glutamine and 1% penicillin–streptomycin. The cells were maintained in a humidified incubator at 37 °C in an atmosphere of 5% CO_2_.

MCF-7 and MDA-MB-231 cells were seeded in 96-well plates at a density of 104 cells/well in 200 μL culture medium and allowed to adhere overnight. Next, the media were removed and the cells were incubated 24 h with HDL (2.5; 5 and 10 mg/mL protein) or Non-HDL (2.5; 5 and 10 mg/mL protein) diluted with serum-free medium containing 0.1% BSA.

To assess the cell viability, MTT solution (50 μL of 2.5 mg/mL) was added to each well to obtain formazan crystals. After 4 h of incubation, the liquid in the wells was removed and t he formazan deposits were solubilized in 200 μL DMSO, after which the absorbance was measured at 570 nm using SpectraMax 190 equipment (VWR, Fontenay sous Bois, France). Cell viability was calculated as percentage of the untreated cells (control).

### 2.4. Statistical Analysis

Statistical analysis was performed on SAS software, version 9.3 (Chapell Hill, NC, USA). Median and distribution (25th–75th) of the variables were estimated. The differences in lipoprotein EPA and DHA between patients with high Ki-67 tumor (Prolif+) and patients with a low Ki-67 tumor (Prolif−) were established by Wilcoxon test in each sub-group of patients (HR+ and HR−).

## 3. Results

### 3.1. Description of Cohort

#### Anthropometric and Clinical Characteristics of the Patients

Patient characteristics are presented in Table 1, based on tumor expression of estrogen/progesterone receptors (HR− and HR+). Within each group, patients were dichotomized according to Ki-67 tumor expression levels ≤ 20% or ≥30%, as low proliferative (Prolif−) and high proliferative (Prolif+), respectively.

Overall, regardless of the hormonal status of the tumor (HR+ or HR−), patients with the least proliferative tumors were older (median [25th–75th]: 64.1 [56.0–72.5] vs. 56.2 [48.0–66.0] years old; *p* = 0.003 for HR+/Prolif− vs. HR+/Prolif−, and 62.5 [56.0–69.0] vs. 54.4 [43.0–66.0] years old; *p* = 0.08, HR−/Prolif− vs. HR−/Prolif−, respectively. Therefore, these women were mostly menopausal. The groups are comparable according to BMI and drug therapy affecting lipid metabolism.

The invasive carcinoma of no special type was the main type of breast cancer diagnosed in HR+ or HR− groups with at least more than 84%, mainly HER2−, except in HR+/Prolif+. As expected, the subgroups of patients differed according to the histopronostic grades with a good prognostic grades (grade I + II) in HR−/Prolif− and HR+/Prolif− with 89% and 92.2%, respectively.

### 3.2. Circulating Lipid Parameters

#### Cholesterol and Triglycerides

Circulating lipid concentrations in each sub-group of patients are summarized in Table 2. Whatever the hormonal status and the level of Ki-67 expression of breast tumor, the lipid parameters of the patients were comparable and close to normal range.

EPA serum concentrations were similar in the four groups of patients (Table 3) with circulating concentration ranging as median [25th–75th] mmol/L, between 1.01 [0.77–1.44] to 1.27 [0.78–1.52] mmol/L for HR−/Prolif+ and HR+/Prolif+, respectively.

However, differences were observed, when considering the lipoproteins carrying the FA, especially in HR− subgroup. Indeed, for HR− status, Non-HDL EPA differed according to Prolif+ and Prolif−, with 0.18 [0.13–0.40] vs. 0.05 [0.02–0.07] mmol/L. The ratio of on-HDL EPA/Apo B and the ratio of Non-HDL EPA/HDL EPA were calculated to evaluate the enrichment of FA per apoB and the enrichment FA carried by apoB-containing lipoproteins (Non-HDL) compared to apoA-I-containining lipoproteins (HDL). These two ratios were significantly higher in Prolif− vs. Prolif+ (*p* < 0.001), 0.12 [0.09–0.18] vs. 0.02 [0.01–0.05] and 0.20 [0.15–0.36] vs. 0.04 [0.02–0.08], respectively.

In the same way, DHA plasma concentrations were similar in the different groups and ranged from 26.90 [19.00–35.90] to 29.45 [18.98–36.45] mmol/L for HR−/Prolif+ and HR−/Prolif−, respectively. There was no difference between the two groups of patients, neither for plasma DHA carried by HDL nor Non-HDL. However, for HR− status a significantly higher molar ratio of Non-HDL DHA/Apo B was observed in HR−/Prolif−, vs. HR−/Prolif+, (*p* = 0.04) with 1.00 [0.73–1.69] vs. 0.52 [0.14–1.08], respectively.

### 3.3. In Vitro Proliferation Studies

The proliferative effects of isolated lipoproteins containing EPA and DHA, on two breast cancer cell line models, were studied by MTT cytotoxicity assay. The Figure 1 illustrates the results obtained for the fraction Non-HDL EPA and Non-HDL DHA on MDA-MB-231, a non-hormone-dependent breast cancer cell line and MCF-7, a hormone dependent breast cancer cell line. Table 4 gives the correlation coefficients observed between lipoprotein FA content and MCF7 and MDA cell proliferation. Proliferation was significantly reduced by Non-HDL DHA and, tended to be reduced by Non-HDL EPA, in MDA-MB-231 cells only.

The correlation between lipoproteins, omega-3 fatty acids carried by lipoproteins and in vitro proliferation has been established (Table 4). Among all the studied fractions, only the correlation between the EPA concentration of Non-HDL was confirmed in vitro, although with borderline statistical significance (*p* = 0.07), in MDA-MB-231 cells. Moreover, Non-HDL DHA, in the same cells model was significantly correlated to proliferation (*p* = 0.04).

## 4. Discussion

Our study was conducted to clarify the role of DHA and EPA, two main nutritional PUFA of marine origin, on breast cancer, with the hypothesis that their impact differed according to the lipoproteins carrying them.

Polyunsaturated fatty acids (PUFA) are composed of two main families, omega 6 and omega 3. The metabolic precursors of these two families are the essential fatty acids linoleic acid (C18:2n-6, LA) and α-linolenic acid (C18:3 n-3, ALA), whose metabolisms lead to the synthesis of long-chain derivatives, especially arachidonic acid (AA) for omega 6 and eicosapentaenoic acid (EPA) and docosahexaenoic acid (DHA) for omega 3, naturally found in fish oil. These PUFA have different functional, energetic and structural physiological roles, and play a role in the prevention of cardiovascular disease and in cognitive development. The role of omega 3, specifically DHA, in brain development and vision justifies supplementation in infant formula [3].

Clinical data exploring the link between ω3 and breast cancer are numerous but remain controversial. A recent work suggests a weak association between ω3 fatty acid intake and breast cancer risk in cohort studies, but no statistically significant association in case-control studies were described [7]. However, observational studies suggest a decrease of breast cancer risk associated with ω3 intake only in obese women [21]. As the association between ω3 PUFA and cancer risk remains controversial, Lee et al. 2020 [7] performed an umbrella review of meta-analyses to evaluate the evidence for the association especially between ω3 fatty-acid intake and breast cancer outcomes. The main results are presented in Table 5.

Weak evidence of an association was shown with *n* = 3 of 14 statistically significant results. Unfortunately, the hormonal status of breast cancer in these studies was not available

The molecular mechanisms underlying these antiproliferative effects have been studied in breast location but also in colorectal cancer [22]. The ability to modify cellular cells by integrating FA in the bi-layer phospholipid membranes inducing modification of fluidity and lipid raft functionality has been strongly suggested [23]. These modifications could modify signal transduction inducing apoptosis. In addition, ω3 PUFA could perform their anti-carcinogenic effects through their metabolites such as resolvines [17,24].

Our studied cohort exhibited the clinico-biological characteristics close to the French population BC patients, as described in a previous published work [17]. Regardless of the type of cancer considered, hormone-dependent or not, the youngest patients exhibited the most proliferative tumors, in agreement with the data in the literature. The % expression of Ki-67 allows researchers to define the proliferation index of the tumors [25], and its use in BC with a cut-off varying according to the authors from 1% to 28% [26]. In order to overcome the lack of consensus concerning this cut-off, we differentiated Ki-67 ≤ 20% as Prolif− and Ki-67 ≥ 30% as Prolif+.

Circulating lipids were similar to those of BC patients in the control arm of a newly diagnosed EPA/DHA supplementation study [27]. In the same way, Non-HDL and total cholesterol values were comparable to those described in a general population of age-matched women [28]. Using circulating lipids as predictive markers of BC is controversial. Thus, pre-operative serum levels of TG and HDL-C may be independent factors to predict outcomes in BC patients [24]. Conversely, some authors suggest a positive association between HDL-C and BC risk in women with extensive mammographic density [22]. However, as described by Maran et al. [27] generally a significant association is strongly suggested between LDL-C and carcinogenesis and oxidized-LDL and metastasis.

Regardless of the hormonal status of the tumors and the proliferative profile, the global circulating concentrations of ω3-PUFA were similar. However, there were differences according to the type of EPA and DHA carrying lipoproteins. Indeed, the fraction of EPA transported by Non-HDL lipoproteins, i.e., LDL, VLDL and IDL, was significantly higher in Prolif− than in Prolif+ patients, and this, only for non-hormone sensitive tumors.

This increase in Non-HDL EPA was due both to an overall increase in the number of carrier lipoproteins, but also to an enrichment in EPA per Non-HDL lipoprotein particle, as suggested by the Non-HDL EPA and the Non-HDL EPA/apoB ratio, respectively. Moreover, this enrichment was at the expense of HDL lipoproteins, as shown by the evolution of the Non-HDL EPA/HDL EPA ratio.

For DHA, we observed only a significant enrichment in the FA per Non-HDL lipoprotein, as shown by the Non-HDL EPA/apo B ratio and the Non-HDL DHA/HDL DHA ratio.

Therefore, a specific protective effect of EPA carried by Non-HDL particles in this tumour subtype is suggested. The differences observed between DHA and EPA remain unclear. Both PUFAs are metabolites of alpha linolenic acid and EPA is essentially derived from the metabolism of PUFA precursors. Retroconversion of DHA to EPA is unlikely, and has only been described in the literature in the case of major dietary intake of DHA [26].

In order to confirm these clinical results, in vitro experiments were performed to evaluate the ability of PUFA lipoprotein fractions to induce, or inhibit apoptosis, on two different cellular breast cancer models, one representative of HR+ tumors (MCF7) and the other (MDA-MB-231) as a model of HR− tumors.

As described in Table 5, in vitro experiments confirmed our clinical data with apoptotic effects of some lipoprotein fractions containing ω3-PUFA in non-hormonal model. Indeed, Non-HDL fractions incubated with MDA-MB-231 cells inhibited cell proliferation in correlation with their EPA and DHA content, thus confirming the clinical observations.

The fact that Non-HDL DHA fractions were significantly and negatively associated with proliferation in the cell model but not in the clinical study remains unclear. However, it should be noted that although the absolute concentration in Non-HDL DHA was not significantly different between high and low levels of Ki-67 (*p* = 0.14), a relative enrichment of these lipoproteins in DHA, as assessed by the Non-HDL DHA/Non-HDL apo B ratio was observed (*p* = 0.04). These in vitro data could be explained by a distinct metabolism between the two models. Indeed, MDA-MB-231 can accumulate more lipid droplets than in the estrogen receptor-positive cell line (MCF7). Sobot et al. [28] found a high expression of LDL receptors in the TNBC cells allowing LDL uptake [12]. Therefore, the pro-apoptotic effects of Non-HDL EPA and Non-HDL DHA could be explained, in part by an increase of their uptake by the cells.

Molecular mechanisms have been explored to explain these effects. DHA induces apoptosis in MCF-7 breast cancer cells in vitro, probably via caspase-8 activation [29]. DHA also seems to be able to decrease the expression of microRNA21 inducing therefore a decrease in growth and proliferation of BC cells [30]. Moreover, DHA appears to be a more potent inhibitor of BC metastasis than EPA [31]. The anticancer effects of these ω3 PUFA have been also demonstrated in vivo. Dietary supplementation with EPA/DHA could contribute to reinforcement of the response to anticancer treatments (chemotherapy and immunotherapy) as in TNBC [32], and also in the basal-like model [33].

Thus, the effects of EPA and DHA on the severity of breast cancer depends on the lipoprotein carrier and these FA have an effect on the severity only in the case of HR− tumors as summarized in the Figure 2.

We have previously shown that the apolipoprotein composition of HDL and Non-HDL may influence the severity of the disease [12]. Therefore, the effect of FA carried by different lipoproteins, as shown in the present study may also depend on the apolipoprotein composition of the lipoproteins. Further studies are necessary to study this potential interaction.

In this study we did not examine the actors involved in proliferation and apoptosis signaling. Several proliferation signaling pathways could be tested, namely those frequently studied with omega-3 fatty acids (MAPK and AKT pathways) [34,35]. The effect of these lipoproteins (HDL and non-HDL) on the lipid rafts of these cancer cells can also be considered since these lipid domains are rich in signaling proteins (e.g., EGFR receptor). EGFR is involved in the activation of the MAPK and AKT signaling pathways. Regarding apoptosis, several actors (Caspases, Bax, PARP) [36] have been affected by omega-3 fatty acids and could be involved in our study. However, other pathways and actors could be involved since HDL and non-HDL carrying omega-3 fatty acids are complex and could also modify cell metabolism.

## 5. Conclusions

This preliminary study suggests a specific protective effect of some long-chain ω3 PUFA, EPA and DHA when carried by Non-HDL lipoproteins, only in patients with hormone-receptor-negative tumors. These results have been observed in a clinical study and confirmed by an in vitro approach. Experiments are currently underway on large cohorts of breast cancer patients to confirm our hypotheses.

## Figures and Tables

**Figure 1 nutrients-14-02461-f001:**
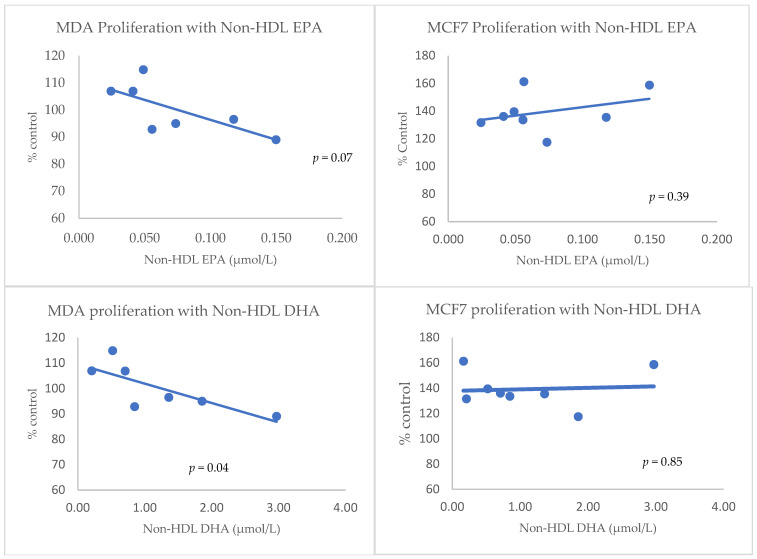
In vitro proliferation assay with different concentrations of Non-HDL EPA (**top**) and Non-HDL DHA (**bottom**) fractions by MTT assay on MDA-MB-231 cells and on MCF-7 cells. Expressed as % for control. HDL: high-density lipoprotein; EPA: eicosapentaenoic; DHA: docosahexaenoic.

**Figure 2 nutrients-14-02461-f002:**
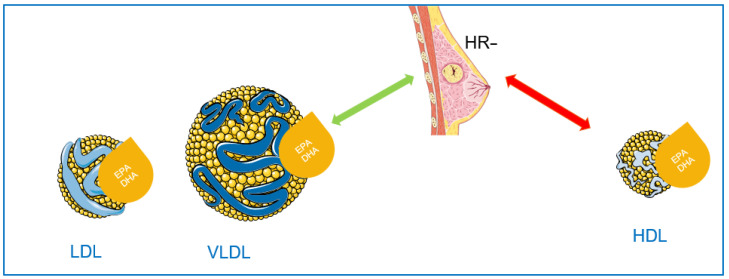
Schematic hypothesis of dual effects of EPA and DHA on HR− breast cancer tumor according to the lipoprotein carriers, green arrow suggesting protective effect compared to red arrow suggesting a pejorative effect. LDL: low-density lipoprotein; VLDL: very low density lipoprotein.

**Table 1 nutrients-14-02461-t001:** Clinicobiological parameters of the studied cohort.

	HR−			HR+		
	Ki-67 ≤ 20% (Prolif−) (*n* = 19)	Ki-67 ≥ 30% (Prolif+) (*n* = 29)	*p*	Ki-67 ≤ 20% (Prolif−) (*n* = 51)	Ki-67 ≥ 30% (Prolif+) (*n* = 41)	*p*
Age (years) *	62.5 [56.0–69.0]	54.4 [43.0–66.0]	0.08	64.1 [56.0–72.5]	56.2 [48.0–66.0]	0.003
BMI (kg m^−2^) *	24.7 [21.8–27.3]	25.2 [22.2–27.1]	0.66	25.4 [22.7–27.5]	26.1 [21.9–28.8]	0.67
Menopausal status **	17 (89%)	21 (72%)	0.15	43 (84.3%)	24 (58.8%)	0.006
Normolipidemic treatments	3 (16%)	4 (14%)	0.85	5 (9.8%)	5 (12%)	0.71
Type of Cancer **			0.29			0.01
Invasive Carcinoma of No Special Type (Ductal)	16 (84%)	28 (97%)		51 (100%)	36 (88%)	
Invasive Lobular Carcinoma	2 (11%)	1 (3%)		0	5 (12%)	
Histoprognostic Grade **			<0.001			<0.001
Grade I	0	0		21 (41.2%)	0	
Grade II	17 (89%)	3 (10.3%)		26 (51%)	6 (14.6%)	
Grade III	2 (11%)	26 (89.7%)		4 (7.8%)	35 (85.4%)	
HER2+	3 (15.8%)	7 (24%)	0.48	5 (9.8%)	14 (34.1%)	0.12

* Median (25th–75th) ** Frequency expressed as *n* (percentage).

**Table 2 nutrients-14-02461-t002:** Circulating lipids (mmol/L) according to the subgroups of patients.

	HR−			HR+		
	Ki-67 ≤ 20% (Prolif−) (*n* = 19)	Ki-67 ≥ 30% (Prolif+) (*n* = 29)	*p*	Ki-67 ≤ 20% (Prolif−) (*n* = 51)	Ki-67 ≥ 30% (Prolif+) (*n* = 41)	*p*
Plasma Cholesterol	5.30 [4.53–5.93]	5.25 [4.07–5.75]	0.77	5.03 [4.17–5.83]	5.36 [4.44–5.91]	0.30
Plasma Triglycerides	1.23 [0.91–1.74]	1.11 [0.79–1.37]	0.38	0.90 [0.71–125]	0.92 [0.74–1.12]	0.83
HDL Cholesterol	1.48 [1.10–1.66]	1.39 [1.07–1.51]	0.14	1.30 [1.09–1.62]	1.38 [1.14–1.57]	0.30
Non-HDL Cholesterol	3.90 [3.19–4.65]	3.79 [2.82–4.46]	0.77	3.57 [2.94–4.39]	3.84 [3.17–4.50]	0.53

Expressed as median mmol/L (25th–75th). *p* value ≤ 0.05, significant difference between the two groups, Wilcoxon test. HDL: high-density lipoprotein.

**Table 3 nutrients-14-02461-t003:** Plasma and lipoprotein fatty acids (mmol/L) and molar ratios with apo A-I (HDL) or apo B (Non-HDL) according to the subgroups of patients.

	HR−			HR+		
	Ki-67 ≤ 20% (*n* = 19)	Ki-67 ≥ 30% (*n* = 29)	*p*	Ki-67 ≤ 20% (*n* = 51)	Ki-67 ≥ 30% (*n* = 41)	*p*
Plasma EPA	1.09 [0.85–1.83]	1.01 [0.77–1.44]	0.77	1.10 [0.84–1.44]	1.27 [0.78–1.52]	0.30
HDL EPA	0.92 [0.67–1.35]	0.98 [0.75–1.42]	0.38	0.85 [0.72–1.42]	1.00 [0.69–1.33]	0.53
HDL EPA/Apo AI	0.019 [0.015–0.023]	0.018 [0.013–0.024]	0.77	0.018 [0.013–0.022]	0.019 [0.014–0.024]	0.53
Non-HDL EPA	0.18 [0.13–0.40]	0.05 [0.02–0.07]	0.0001	0.09 [0.06–0.17]	0.1 [0.06–0.31]	0.53
Non-HDL EPA/Apo B	0.12 [0.09–0.18]	0.02 [0.01–0.05]	0.0001	0.05 [0.02–0.10]	0.075 [0.031–0.161]	0.53
Non-HDL EPA/HDL EPA	0.20 [0.15–0.36]	0.04 [0.02–0.08]	0.0001	0.10 [0.05–0.20]	0.13 [0.05–0.27]	0.53
Plasma DHA	29.45 [18.98–36.45]	26.90 [19.00–35.90]	0.77	27.76 [21.64–34.04]	27.15 [20.58–35.56]	0.83
HDL DHA	27.07 [17.72–34.03]	26.70 [17.30–35.10]	0.77	26.13 [20.29–32.35]	23.53 [19.42–33.82]	0.83
HDL DHA/Apo AI	0.50 [0.35–0.73]	0.45 [0.34–0.61]	0.38	0.46 [0.38–0.64]	0.54 [0.37–0.63]	0.30
Non-HDL DHA	1.74 [1.09–2.43]	1.08 [0.30–2.04]	0.14	0.88 [0.45–1.50]	1.31 [0.52–2.46]	0.14
Non-HDL DHA/Apo B	1.00 [0.73–1.69]	0.52 [0.14–1.08]	0.04	0.52 [0.22–0.82]	0.70 [0.37–1.21]	0.06
Non-HDL DHA/HDL DHA	0.08 [0.04–0.10]	0.04 [0.01–0.07]	0.04	0.04 [0.02–0.06]	0.05 [0.02–0.08]	0.30

Expressed as median [25th–75th]. *p* value ≤ 0.05, significant difference between the two groups, Wilcoxon test (indicated in bold). HDL: high-density lipoprotein; EPA: eicosapentaenoic; DHA: docosahexaenoic.

**Table 4 nutrients-14-02461-t004:** Correlation between lipoproteins, omega-3 fatty acids carried by lipoproteins and in vitro proliferation.

	MDA-MB-231 Cells		MCF-7 Cells	
	β ± s.d.	*p*	β ± s.d.	*p*
HDL Cholesterol	2.16 ± 8.10	0.80	−3.07 ± 12.22	0.81
HDL EPA	−10.14 ± 9.18	0.32	9.46 ± 15.30	0.56
HDL EPA/HDL Apo AI	−523.92 ± 550.22	0.38	555.41 ± 889.04	0.56
HDL DHA	−0.24 ± 0.49	0.65	0.80 ± 0.70	0.30
HDL DHA/HDL Apo AI	6.80 ± 27.21	0.81	39.27 ± 38.90	0.35
LDL Cholesterol	2.78 ± 3.53	0.47	−8.21 ± 4.36	0.11
Non-HDL EPA	−147.47 ± 65.56	0.07	121.88 ± 130.59	0.39
Non-HDL DHA	−7.62 ± 2.80	0.04	1.20 ± 6.15	0.85

**Table 5 nutrients-14-02461-t005:** Meta-analyses of fish and *ω*3 fatty-acid intake and breast cancer risk according to re-analyses of [7].

Type of Studies	Type of ω-3 Fatty Acid Intake	Number of Cases	Relative Risk 50% CI	*p* Value
Nest CC, CC, cohort	Highest marine n-3 PUFA intake	16,178	0.86 (0.78, 0.94)	0.002
CC, cohort	Marine n-3 PUFA (Diet)	11,519	0.86 (0.76, 0.96)	0.007
Cohort	Per 0.1 g/d increment of dietary marine n-3 PUFA	3114	0.93 (0.90, 0.97)	3.89 × 10^−4^
CC, cohort	Total n-3 PUFA	NR	0.96 (0.86, 1.07)	0.43
Cohort	Per 0.1% energy increment of daily dietary marine n-3 PUFA	6344/288,626	0.97 (0.92, 1.02)	0.22
CC, cohort	Highest dietary fish intake	13,323/687,770	1.03 (0.93, 1.14)	0.61
CC, cohort	Per 15 g/d increment of fish intake	13,323/666,400	1.00 (0.97, 1.03)	0.98
CC, cohort	Marine n-3 fatty (EPA)	NR	0.86 (0.75–1.01)	0.098
CC, cohort	Marine n-3 fatty (DHA)	NR	0.89 (0.75, 1.05)	0.16
CC, cohort	Marine n-3 fatty (DPA)	4746/284,724	0.91 (0.68, 1.22)	0.54
Cohort	ALA (diet)	8274/281,756	0.98 (0.90–1.06)	0.56
Cohort	Per 0.1 g/d increment of dietary ALA intake	6310/190,451	1.00 (0.99–1.01)	0.54
Cohort	Per 0.1% energy increment of daily dietary ALA intake	5510/171,680	1.00 (0.99, 1.01)	0.96
CC, cohort	ALA (tissue biomarker and diet)	9296/284,724	0.97 (0.90, 1.04)	0.39

CC, case control; NR, not reported; DPA, docosapentaenoic acid; ALA, α-linolenic acid.

## Data Availability

Data are available on request.

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
