# Peer review of "Link between Omega 3 Fatty Acids Carried by Lipoproteins and Breast Cancer Severity"

_nutrients, 2022, doi:10.3390/nu14122461_

Round 1

Reviewer 1 Report

The paper entitled "Link between omega 3 fatty acids carried by lipoproteins and 2 breast cancer severity" has explored the association between Dieterty Omega3 and breast cancer using in vitro and in vivo data. Most of the comments are related to the discussion part.

1- Page 7, lines 226 to 234: move this section to the introduction. You are not discussing anything here. Please keep the discussion concise and go directly to the point.

2- Page 7, line 242: reference needed for the sentence finishing with  "... but also clinical studies." You may refer to the table suggested below.

3- Page 7, lines 244-248: Please make a table and cite the most critical and relevant findings from other papers. This is an essential table, as the subject is really controversial. You may add the number of patients, status (Her2+ or -), type of Omega3 fatty acid (if known), and the study outcome.

4- In the discussion, please clearly mention the limitations of this study.

5- This paper needs an illustration. Please draw your idea and suggested mechanism in a simple form. It will help other people better understand what you refer to in your paper. It is also suitable for educational purposes. You may include it somewhere in the discussion. Adding too much text and statistical tables is not professional.  

6- There are also typographical errors (some of them are ugly such as using RH instead of HR on the first page). Please be mindful of these sorts of minor (but significant) errors.

Stay safe, and sorry for the delay.

Reviewer 2 Report

This is an interesting paper to explore the link between omega 3 fatty acids and breast cancer severity.  The data is very preliminary, however, it found Non-HDL EPA differentiated between Ki-67 <20% and >30% only in patients with hormone receptor negative tumors.  The approaches of the LCMS work and the ELISA quantifications are robust. However, the functional follow up with cell line in vitro proliferation studies is a stretch to validate the biology.  The manuscript should be cautious in interpreting the cell line results as a validaiton.

Reviewer 3 Report

This is an original work addressing a topic of interest, aiming to assess whether omega 3 supplementation impairs the severity of breast cancer patients. Therefore, I made considerations so that the authors can think about it and seek to improve this study.

  1. Line 284, page 8, is missing a period at the end of the sentence.
  2. Why did the authors not perform assays in cell lines to further investigate the proteins that participate in apoptosis and proliferation? How would the profile of these proteins be in the in vitro study?
  3. After all, is omega 3 supplementation harmful or beneficial in breast cancer patients?
